# A Survey on Robotic Technologies for Forest Firefighting: Applying Drone Swarms to Improve Firefighters' Efficiency and Safety

Juan Jesús Roldán-Gómez [1,*], Eduardo González-Gironda [2] and Antonio Barrientos [2]

1. Departamento de Ingeniería Informática, Escuela Politécnica Superior, Universidad Autónoma de Madrid, Francisco Tomás y Valiente, 11, 28049 Madrid, Spain
2. Centro de Automática y Robótica (UPM-CSIC), Universidad Politécnica de Madrid, José Gutiérrez Abascal, 2, 28006 Madrid, Spain; eduardo.gonzalez.gironda@alumnos.upm.es (E.G.-G.); antonio.barrientos@upm.es (A.B.)
* Correspondence: juan.roldan@uam.es

**Abstract:** Forest firefighting missions encompass multiple tasks related to prevention, surveillance, and extinguishing. This work presents a complete survey of firefighters on the current problems in their work and the potential technological solutions. Additionally, it reviews the efforts performed by the academy and industry to apply different types of robots in the context of firefighting missions. Finally, all this information is used to propose a concept of operation for the comprehensive application of drone swarms in firefighting. The proposed system is a fleet of quadcopters that individually are only able to visit waypoints and use payloads, but collectively can perform tasks of surveillance, mapping, monitoring, etc. Three operator roles are defined, each one with different access to information and functions in the mission: mission commander, team leaders, and team members. These operators take advantage of virtual and augmented reality interfaces to intuitively get the information of the scenario and, in the case of the mission commander, control the drone swarm.

**Keywords:** robotics; multi-robot systems; swarms; drones; firefighting

## 1. Introduction

Forest fires are one of the most common and, at the same time, serious emergencies facing humanity. They threaten not only natural areas, where they cause important losses of plant and animal diversity, but also urban areas, where they can cause dramatic human and material losses. Furthermore, forest fires cause significant emissions of greenhouse gases and consequently are contributing to global warming. For these and other reasons, states develop policies for fire prevention, early detection, and rapid intervention.

Quantifying the fires and their consequences along the world is not a trivial task. According to the World Fire Statistics [1], a report published by the International Association of Fire and Rescue Services that collects data from multiple governments, there were 4.5 million fires and 30,800 deaths in countries with 2700 million inhabitants in 2018, which means 1.7 fires per 1000 and 1.1 deaths per 100,000 inhabitants that year. Although these figures do not take into account the whole world, they allow us to quantify the magnitude of the problem. The information provided by several space agencies supports this thesis: the European Space Agency (ESA) publishes the World Fire Atlas with the information collected by ATSR-2 [2] and Sentinel-3 [3]), whereas the National Aeronautics and Space Administration (NASA) does the same with the Global Fire Atlas [4].

Current forest firefighting missions consider prevention, surveillance, and extinguishing tasks. The first ones seek to prevent the occurrence of fires and limit their consequences, the second ones look for detecting fires early, and the third ones search to put them out quickly and safely. Firefighters reveal the lack of human and material means and the degraded information of the scenario as the main problems in these tasks. They routinely use

multiple types of vehicles and machinery to improve the performance and safety of these operations. However, the use of robots and especially drones is not common, although these autonomous systems could solve some of the current challenges.

This paper aims at analyzing the current problems in forest firefighting missions and the potential of robotic technologies to solve them. Therefore, we pose the following two research questions:

1. What are the main problems in current forest firefighting missions?
2. How can robotic technologies contribute to solving them?

For this purpose, the paper analyzes the data provided by governments, the results of two original surveys on firefighters, and the literature on robotics applied to forest firefighting.

Finally, the paper proposes a concept of operation for the application of drone swarms to fire prevention, surveillance, and extinguishing tasks.

The remainder of the paper is organized as follows: Section 2 addresses the current situation of firefighting, analyzing the public statistics provided by multiple countries and presenting the results of our surveys and interviews to professionals. Section 3 collects various works developed in the context of academy and industry that apply robots to firefighting tasks. The concept of operations using drone swarms to support firefighters in all these tasks is presented in Section 4. Finally, the main conclusions of the work are summarized in Section 5.

## 2. Firefighting State

This section analyzes the current state of firefighting. For this purpose, it describes current operations of fire prevention, surveillance, and extinguishing, collects relevant statistics to identify main problems, and presents the opinions of professionals through two surveys. Note that most of the information presented in this section is from Spain, but can be generalized to at least European and Mediterranean countries.

In 2019, 10,883 fires burned 83,963 ha in Spain: 7290 of these fires affected less than 1 ha, whereas 3593 affected more than 1 ha [5]. As reported, these figures were similar to the previous years, having an average of 12,182 fires and 99,082 ha per year between 2009 and 2018. In other words, every year, 0.356% of the forest surface of Spain suffers the consequences of fires.

Most of these fires occur in spring and summer, especially in March, July, August and September, whereas the worse consequences occur in July and August when more surface burns than the rest of the year [6]. In the case of summer, this behavior can be explained by the high temperatures, which favor the appearance of fires and their spread throughout the territory. In the case of March, most of these fires occur in the north-west of the country and are caused by an accidental, negligent, or intentional use of fire. However, firefighting is performed throughout the year, since it involves not only extinguishing fires but also preventing them.

Fire prevention involves a set of activities that seek to reduce the probability of fire occurrence, as well as to limit their effects if they occur [7]. More than half of forest fires in Spain between 2006 and 2015 were caused intentionally, whereas 28% were caused by accidents or negligent behaviors, 7% had natural origins, and 12% still have unknown causes. Therefore, there are two main groups of activities: prevention on causes and prevention on combustibles. The first category groups those activities that seek to reduce the risks and usually present a social character, such as the awareness campaigns to avoid the use of fire in the primary sector and negligent behaviors in natural environments. The second one covers the actions performed on land uses and vegetation distribution, which seek to generate discontinuities to prevent the expansion of potential fires.

Fire surveillance involves the activities performed to detect fires as early as possible. The damages caused by forest fires highly depend on detection and response times. The information of Spain in 2019 is clear: the average burned surface when response time was shorter than 1 h was 7.10 ha, whereas the one when response time was longer than 1 h was 30.66 ha [5]. For this reason, minimizing detection and response times is key for firefighting.

Currently, detection times are addressed with a watchmen network distributed throughout the land and, to a lesser extent, ground and aerial mobile surveillance. In Spain, 60% of fires are detected thanks to citizen collaboration, 27% by static watchmen, 1.6% by mobile watchmen, and 0.5% by aerial means [6]. Meanwhile, response times are addressed by the effective coordination of the teams and the use of helicopters to deploy firefighters in the affected area.

Fire extinguishing involves not only the actions performed to put out the flames but also some activities that support these actions, such as creating firewalls, routes for entry and exit of vehicles, runways, heliports, etc. In Spain between 2006 and 2015, these activities involved the participation of humans (100% of fires), ground vehicles and machinery (94.8%), and aerial means (23.5%) [6]. Extinguishing operations are dangerous because any accident can cause injuries or even deaths among the professionals. The government of Spain reports 24 accidents between 2006 and 2015 with 37 deaths, including only firefighting professionals [6]. According to this study, the causes of these deaths were air accidents (43%), entrapments (30%), medical problems (8%), falls (8%), accidents with vehicles (5%), and accidents with machinery (5%). Therefore, it would be good if technological solutions could reduce both accident rate and mortality in the cases of entrapments and falls, which can be caused by the lack of information about fire evolution and terrain features.

We performed a set of surveys and interviews with firefighting professionals to check and broaden this information. The surveys allowed us to involve a high number of professionals and distinguish collective consensus from individual opinions. Meanwhile, the interviews were done before and after the surveys: the first ones allowed us to prepare the questions, whereas the second ones provided us with more details about the answers. These activities aimed to collect information about current problems of firefighting and opinions about potential technological solutions.

Two surveys were carried out with firefighting professionals: one focused on their problems at work (see Section 2.1),and another on their opinion about multiple technologies (see Section 2.2). Both surveys had between 10 and 20 questions and required fewer than 5 min to maximize the answer ratio. The separation of problem and technology surveys prevented the influence of the questions of one on the responses of the other.

The dissemination of the surveys sought to reach professionals who perform all the firefighting roles in most of the regions of Spain. For this purpose, we sent the surveys by email to fire stations and firefighter unions, as well as share them in firefighting groups on various social networks. In this way, we avoided getting a sample biased towards a specific firefighting role or geographic area.

*2.1. Problem Survey*

Our first survey was focused on the problems on current forest firefighting missions. We performed this survey to obtain more information about the first research question. Although the official data previously analyzed are useful to answer this question, the opinions of the professionals involved in these activities are also relevant.

In this survey, we pose the following questions:

- Importance of prevention tasks: As previously mentioned, there are two prevention strategies: those focused on causes and those centered on combustibles. This question seeks the importance that professionals give to each one of these strategies.
- Problems in prevention tasks: This question seeks to find the most relevant problems in current prevention activities, according to the opinions of firefighters.
- Importance of surveillance means: As previously pointed out, forest fires can be detected by citizen collaboration, ground watchmen, and aerial means. This question seeks the importance that professionals give to each one of these means.
- Problems in surveillance tasks: This question seeks to find the most relevant problems in current surveillance activities, according to the opinions of firefighters.

- Problems in extinguishing tasks: This question seeks to find the most relevant problems in current extinguishing activities, according to the opinions of firefighters.

We had the support of several firefighting professionals in the writing of the questions and their possible answers. In this way, we could check that our surveys were sound and easy to understand by our target public. Furthermore, we sent the questionnaires to a sample of twenty professionals before their dissemination to check if they could understand them adequately. Finally, we allowed open answers to some questions and shared our contact data to receive doubts.

This survey was sent via email to fire stations, unions, and associations, as well as shared with firefighters' communities on several social networks. A total of 140 professionals from different regions of Spain took part in that survey in three weeks (note that this survey is still open to new responses (Forest fires in Spain: Problem survey (https://forms.gle/e4327HBxqqWVMUbY7) [in Spanish])). A summary of the results is shown in Figure 1.

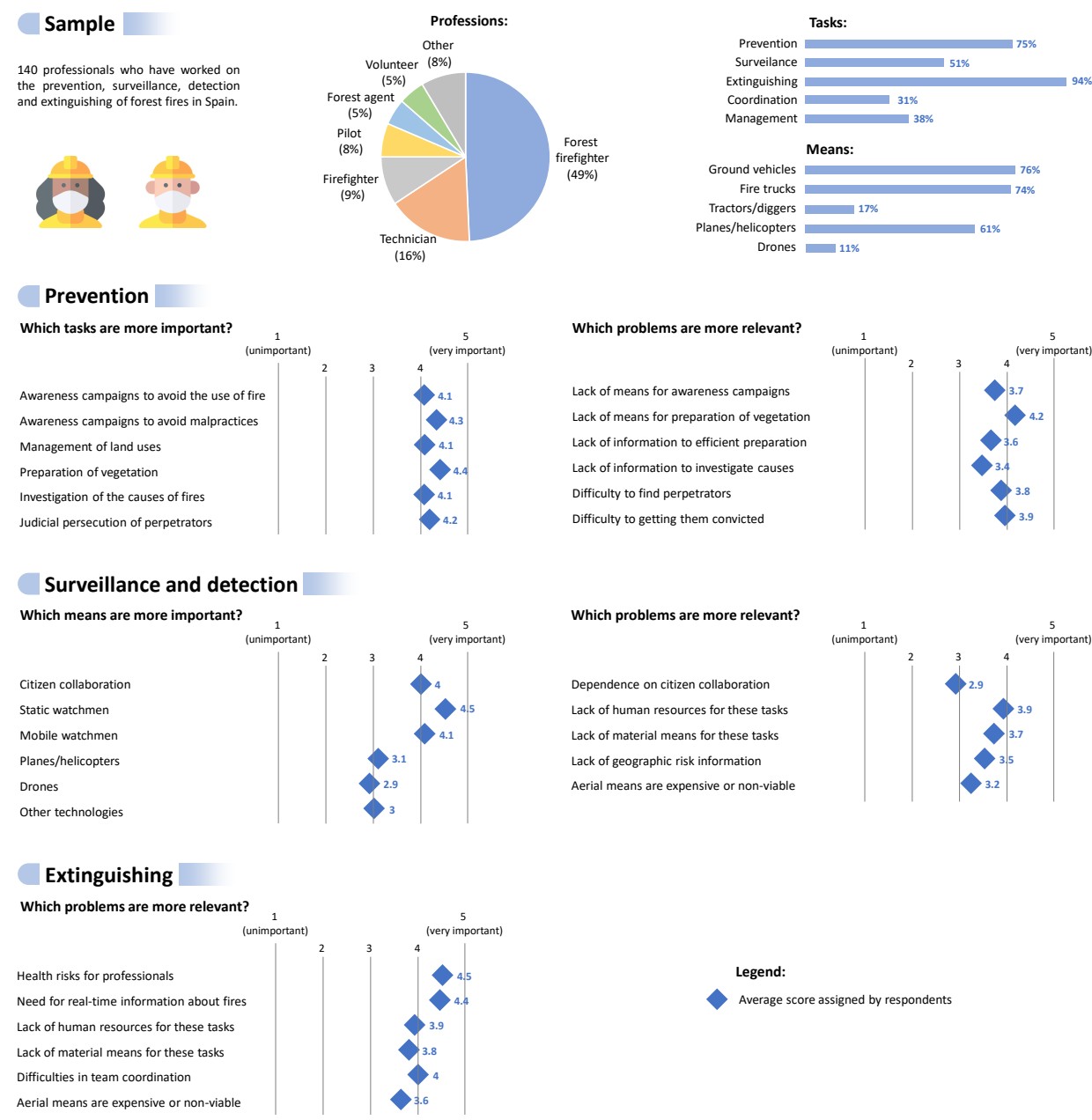

**Figure 1.** Results of the problem survey about forest fires in Spain.

The survey sample is representative of the professionals involved in forest firefighting in Spain, given that it includes not only the main roles (forest firefighters, technicians, firefighters, forest agents, and volunteers) but also other less common roles (pilots, army forces, researchers, support staff, meteorologists...), in comparison to the official reports [5]. The vast majority of them take part in extinguishing tasks (94%), whereas three quarters have experience in prevention and a half in surveillance tasks. Besides these fundamental tasks, around one-third of the respondents have carried out the coordination of operations (31%) and management of human and material resources (38%). Regarding the means used to perform these tasks, three quarters used ground vehicles and fire trucks, sixty percent aerial vehicles (this includes pilots and airborne firefighters), and fewer tractors (17%) and drones (11%).

Spanish firefighters often say that "summer fires should be extinguished in winter", remarking on the importance of prevention activities in firefighting. In this sense, the professionals surveyed assign very close scores to all the prevention tasks. Mainly, they give a slightly higher score to the preparation of vegetation (4.4 in a scale from 1 to 5), and prioritize awareness campaigns against malpractices (e.g., barbecue, smoke or throwing glass bottles in the bush) over campaigns against the use of fire in agricultural and livestock activities (4.3 vs. 4.1, respectively). There are more differences between the main problems faced in prevention tasks. They highlight the lack of means for preparing the vegetation (4.2), together with the difficulty to find perpetrators (3.8) and convict them (3.9).

Regarding surveillance and detection tasks, the participants highlight static watchmen (4.5), mobile watchmen (4.1), and citizen collaboration (4). Note that this evaluation does not coincide with the actual situation, given that sixty percent of fires are detected by citizen collaboration, whereas only twenty-six percent are detected by static watchmen and less than two percent by mobile watchmen. Aerial means are considered less important for fire surveillance and detection: planes and helicopters receive 3.1 points and drones 2.9 points. According to their opinions, the main problem in these tasks is the lack of human resources (3.9), followed by the lack of material means (3.7), and the lack of risk information, which would allow reinforcing surveillance in areas with a higher risk of fire.

Finally, the professionals surveyed consider health risks and the need for real-time information as the main problems in extinguishing tasks with 4.5 and 4.4 points, respectively. Both problems are closely related, considering that most accidents are caused by the lack of information about the fire evolution, such as entrapments and falls. Other relevant problems are the lack of human and material resources (3.9 and 3.8, respectively), and the difficulties to coordinate the teams on the ground (4).

### 2.2. Technology Survey

Our second survey was focused on some technologies that can contribute to solving the reported problems. In this case, we performed this survey to obtain more information about the second research question. The objective was to collect opinions from professionals to estimate their predisposition to use these technologies.

For this purpose, the survey included the target technologies of this study (drone swarms and immersive interfaces), together with some control technologies. These technologies were chosen after a review of research and the commercial literature and served as a reference in the evaluation of target technologies. Incentive systems

- Prevention: The survey considers a solution of prevention on causes (incentive systems for farmers/ranchers to prevent their use of fire) and two solutions of prevention on combustibles (drone and satellite images to support the preparation of vegetation). In this way, two comparisons can be performed: one among the two strategies for prevention, and another between the two technologies that support the vegetation preparation.
- Surveillance: The survey considers two detection systems: one with drones and another with fixed cameras. In this way, the target technology can be compared with a well-known and widely-used surveillance system. Additionally, it includes the use

of artificial intelligence to predict the risk of fire, which allows performing this task over specific areas.

- Extinguishing: The survey asks about the application of drones to monitor the evolution of fires. In addition, it considers three alternatives to receive the information during field operations: an immersive interface, a mobile device, and a voice assistant. In this way, the target technology can be compared to two common methods to receive information.

We took the same measures as in the previous survey to ensure that questions and possible answers were understandable.

The participants of the first survey who gave their emails were invited to fill the second survey. In this case, a total of 70 professionals submitted their responses in the first three weeks (again, the survey is still open to new responses (Forest fires in Spain: Technology survey (https://forms.gle/XV4ScxL9jyCgr4fJ7) [in Spanish])). A summary of the results is shown in Figure 2.

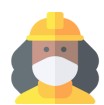 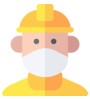 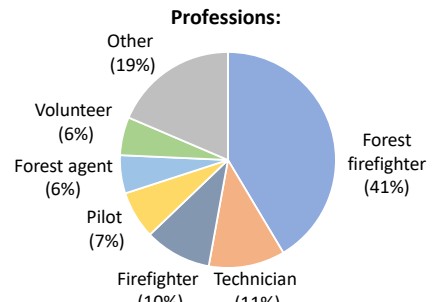 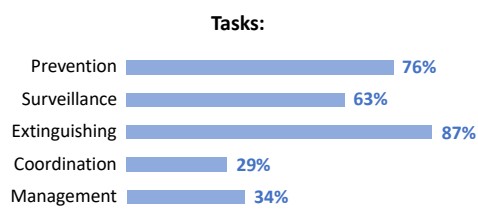
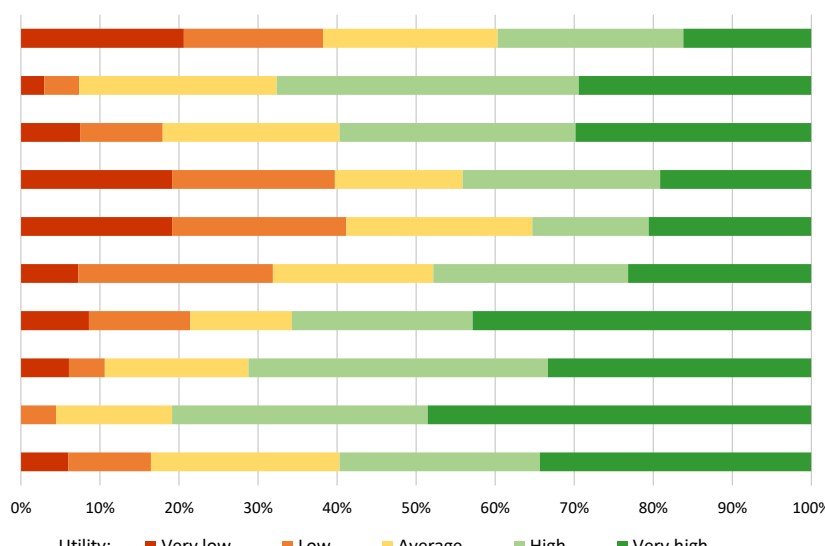

**Figure 2.** Results of the technology survey about forest fires in Spain.

As seen in Figures 1 and 2, the survey samples are very similar, only presenting small variations in the professions and tasks. The professionals could evaluate the utility of the different technologies with five ratings: "very high", "high", "average", "low", and "very low". However, we analyze the results considering three evaluations: positive (including "very high" and "high" answers), neutral (equivalent to "average" answer), and negative (including "low" and "very low" answers).

Three technological solutions for supporting prevention were evaluated: a system with incentives to avoid the use of fire in primary sector activities, satellite images to support the preparation of vegetation, and drone images for the same purpose. In this case, the professionals surveyed evaluate more positively the use of satellite and drone images (approximately, 60–70% positive, 20% neutral, and 10–20% negative).

Another three technological solutions were presented for surveillance and detection tasks: a system with cameras to monitor large and/or remote areas, autonomous drones to cover hard-to-reach areas, and the use of artificial intelligence to predict the risk of fire in every location. This last system received the best rating with 48% positive, 20% neutral, and 32% negative, whereas the other two received ratings of 35–44% positive, 16–24% neutral, and 40–41% negative.

A combination of two technologies was considered for extinguishing tasks: a first one for collecting real-time data from the fire, and a second one to display this data to the firefighter teams. For the first, we presented a fleet of autonomous drones that can monitor fire evolution in real-time. This solution is one of the best valued and the best that uses drones, having 66% positive, 12% neutral, and 22% negative evaluations. For the second, we posed three alternatives: augmented reality headset, mobile device, and voice assistant. The three alternatives are evaluated positively, but the mobile device receives the best score: 81% positive, 15% neutral, and 4% negative. Augmented reality receives 71% positive, 18% neutral, and 11% negative, whereas voice assistant achieves 60% positive, 23% neutral, and 17% negative. These results reveal that participants prefer visual over aural feedback and well-known over new devices, as well as they do not matter using their hands to manage these devices during operations.

A certain bias was detected in the evaluations of technologies by professionals, which positively affects those that are presented as a support for their current operations and negatively those that can change those operations or even threaten their jobs. This can be seen in detection, where they think that artificial intelligence can improve their effectiveness in surveillance, while they feel more threatened by autonomous systems with drones or cameras. Furthermore, the preference for well-known systems for receiving information (mobile devices) over more innovative ones (augmented reality) also reveals this conservative bias.

## 3. Firefighting Robots

Once we have analyzed the current state of firefighting and the opinions of professionals, we must address a new question: "can technology help to solve any of the presented problems?" This section collects the most relevant works that apply robotic and automation technologies to firefighting activities. Our analysis focuses on multi-robot systems and aerial robots used for the prevention, surveillance, and extinguishing of forest fires. However, relevant works that propose other types of robots and consider urban or indoor scenarios are also featured.

As previously occurred with industry, agriculture, and services, robots are being applied to intervene in emergencies and, more specifically, to fight against fires. According to our survey, firefighters are receptive to these technologies when they support their work and do not change its conditions. A previous study with fire chiefs of New Jersey (United States of America) supports these conclusions: they are willing to use drones in firefighting operations, but they point out budget, manpower, and regulation issues [8]. The public opinion about the use of drones for cargo, passenger, and commercial transportation is analyzed by [9], including explicitly firefighting in this last group of applications. The participants of this study support the use of drones for cargo and commercial applications, but they prefer piloted aircrafts for passenger transportation. Finally, a comprehensive survey on the public opinion about drones considering multiple applications and risks can be found in [10].

As most of the relevant articles focus on one or a few specific tasks, we have classified them into prevention (Section 3.1), surveillance (Section 3.2), and extinguishing

(Section 3.3). This classification is supported by several papers in the literature: for instance, Ref. [11] distinguishes between activities before fire (vegetation mapping, surveillance, and risk estimation), during fire (detection and extinguishing), and after fire (ember search and damage assessment).

### 3.1. Prevention

As already explained, prevention is considered the first step of firefighting and encompasses two classes of actions. The first ones involve social activities that seek to prevent fires from occurring, usually developing awareness campaigns targeting key groups as farmers or tourists. The second ones group multiple works on vegetation to reduce the risk of fire and generate discontinuities to difficult their propagation. Logically, the potential of robots to improve current results is higher in these latter operations. The preparation of vegetation is an activity that requires remarkable efforts, where a lack of human and material resources is perceived. Robotic technologies can make this activity more efficient in two ways.

On the one hand, drones can take aerial images that can be used to plan these tasks: detecting the most problematic areas, selecting the vegetation to remove, planning routes for its extraction, etc. Some techniques developed for precision agriculture can be applied in this context [12], such as the detection and identification of plants and trees in high-resolution images [13], three-dimensional LIDAR scans [14], and multispectral images [15] acquired by drones. In all these tasks, drones have been revealed as a suitable alternative to satellites, since they offer greater availability at a lower cost, as well as they are less dependent on the weather conditions in the area of interest [16].

On the other hand, ground robots can support the activities aimed at remove vegetation in forests, playing an intermediate role between the manual labor of firefighters and the heavy machinery used by them. These robots can reach a compromise between the flexibility and precision of firefighters and the quickness and performance of machinery. Forestry and agricultural robots share some challenges and requirements [17], such as the locomotion in rough terrains, localization and mapping in unstructured environments, and planning under uncertainty [18].

A comprehensive fire prevention solution is being developed in the SEMFIRE Project [19], which proposes a multi-robot system to reduce the fuel accumulation in forests and assist in landscaping maintenance. This system consists of small flying robots for vegetation mapping and large-sized tracked mobile robots for forestry mulching.

### 3.2. Surveillance

Fire surveillance is the most covered activity in the literature about robotics for firefighting. Most of the proposals involve the use of different kinds of aerial robots (fixed-wing and multi-rotor drones) equipped with various types of cameras (RGB, infrared, multispectral...) to watch over the forests from above. Fire surveillance tasks may have up to four objectives: search of potential fires, detection to alert firefighters, diagnosis to get relevant data about the fire, and prognosis to predict fire propagation [20]. The early detection of fire is as important as the complete analysis of it, given that firefighting teams need information such as the ignition and danger potential to organize their operations [21].

Unmanned aerial vehicles (UAVs) with on-board vision systems have considerable potential in the detection and monitoring of forest fires, since they offer high maneuverability, flexible perspective and resolution, and limited risks to people [22]. For this purpose, surveillance systems should integrate six elements: a fleet of UAVs with payloads, sensor fusion and image processing methods, guidance, navigation and control (GNC) algorithms, coordination and cooperation strategies, path planning algorithms, and ground control stations (GCS) [23]. The selected UAVs shall meet a set of requirements, such as long flight time, accurate localization with the data obtained by the Inertial Measurement Unit (IMU) and Global Navigation Satellite System (GNSS), stable and robust flight, and good image quality [24].

There are multiple approaches to develop vision systems to detect fires. The work in [25] comprehensively analyzes the potential sensors and methods for terrestrial, aerial, and satellite-based fire detection systems. Regarding the hardware, they use visible [26,27], thermal [28,29], multispectral [30,31] and infrared cameras [20,32], as well as environmental sensors (mostly used in indoor scenarios [33], but also proposed for forests [21]). Regarding the software, traditional computer vision algorithms [22,34] compete with recent artificial intelligence solutions [35,36]. The most common features used to recognize fires in aerial images are color, geometry, and movement. Color and geometry allow detecting potential fires in isolated frames, whereas movement is relevant to check these detections with the whole sequence of frames [22]. A challenge for these algorithms is adapting to different types of fires and scenarios: for instance, subterranean fires show up as columns of smoke, in contrast to common surface forest fires [37].

Heterogeneous multi-robot systems are also considered for fire surveillance. The work presented in [38] proposes an air-ground robotic team, where the Unmanned Ground Vehicles (UGVs) compensate for the weaknesses of UAVs, such as their limitations in autonomy (flight time) and payload (weight capacity). This work proposes the use of UGVs to transport UAVs to the fire scenario, where UAVs can take off, perform their tasks, and land again. Additionally, UGVs are used as base stations for UAVs, centralizing the communications between the fleet, processing the data collected by them, and coordinate their tasks in the scenario. Moreover, the work published in [35] proposes the use of two different types of drones: fixed-wing UAVs for medium-altitude flights searching fires and rotary-wing UAVs for low-altitude flights checking detections. The need for checking detection to avoid false alarms is also expressed in [39], which suggests the use of multiple drones to collect simultaneous information of every area, as well as the use of various features to detect fires in the provided images (e.g., color and movement).

### 3.3. Extinguishing

In general terms, fire extinguishing is the last task of firefighting after having detected and checked the fire. Currently, this task is mostly performed with ground and aerial means that require human intervention. Sometimes, the presence of humans results in risky situations for their lives due to the virulence of fires. Although this is a good reason to try to use robots in these tasks, these autonomous systems are only used experimentally. The literature considers two main approaches: one for aerial extinguishing and another for supporting ground operations.

The main idea of the firefighting drones is to attack the fire when it is in its first stages, trying to avoid the spread of it. An extinguishing quadcopter equipped with a bucket to capture and release water is presented in [40]. Although this design is similar to those used in current firefighting helicopters, the limitation in the payload capacity of the quadcopter reduces its performance.

An aerial hose-type robot that can fly directly into the fire source by a water jet is presented in [41]. This robot receives a continuous intake of water for fighting the fires and controlling its stability. In this way, it solves the limited payload issues of conventional drones, but it requires a water source close to the fire scenario. Another alternative is the use of gases instead of water. A quadcopter that carries a balloon filled with helium is proposed by [42]. This inert gas is used because it can reduce the amount of oxygen of the flames, as well as it is light enough to be transported by a quadcopter. The scalability of this system to forest fires must be validated, including the mechanism to release the balloons on the exact points.

A common idea for putting out fires is the utilization of extinguishing balls [43]. These elements burst when they come into contact with high temperatures, releasing some chemical components that put out the fire. Ref. [44] proposes a quadcopter that can launch an extinguishing ball to the flames of urban and wildfires. Following the same approach, Refs [45,46] propose some alternatives for the release mechanism to allow throwing multiple balls and keep the stability of the drone. Finally, ref. [47] poses a

swarm of UAVs that can perform monitoring and extinguishing tasks, demonstrating the scalability of fire extinguishing systems based on drones that release balls.

Different types of multi-robot systems are proposed for fire extinguishing missions. There is a trend in the literature to apply multiple light robots instead of developing drones with the capabilities of planes and helicopters. For instance, a drone fleet is proposed in [48] and a drone swarm in [49]. When multiple drones work in the same scenario, the coordination of the fleet becomes relevant. The literature contains various proposals of algorithms to allocate targets among the drones, seeking to minimize traveling distance for every drone. Some examples are [11], which proposes that the team shares all the information of the mission and runs an auction-based mechanism to distribute the tasks, and [50], which describes a deep learning method to allocate tasks, overcoming the sensing, communication, and motion limitations of drones.

In addition to fire extinguishing tasks, robots can be used to monitor fires and provide information to firefighters. Ref. [51] describes a novel algorithm for safe human-robot coordination in wildfires. The drones track the evolution of fires, which can be stationary, moving, and moving/spreading, and a human safety module detects if there are humans close to fire spots. Moreover, ref [52] three types of drones to perform patrolling, confirmation, and monitoring tasks, as well as a fire-spreading model to use the information collected from the fires to predict their behavior.

## 4. System Overview

After analyzing the current state of firefighting operations and proposals of robotic systems to perform them, we present a comprehensive concept of operation to apply drone swarms in firefighting missions. This concept of operation is shown in Figure 3 and described in the following subsections: mission in Section 4.1, drone swarm in Section 4.2, team in Section 4.3, and required infrastructure in Section 4.4.

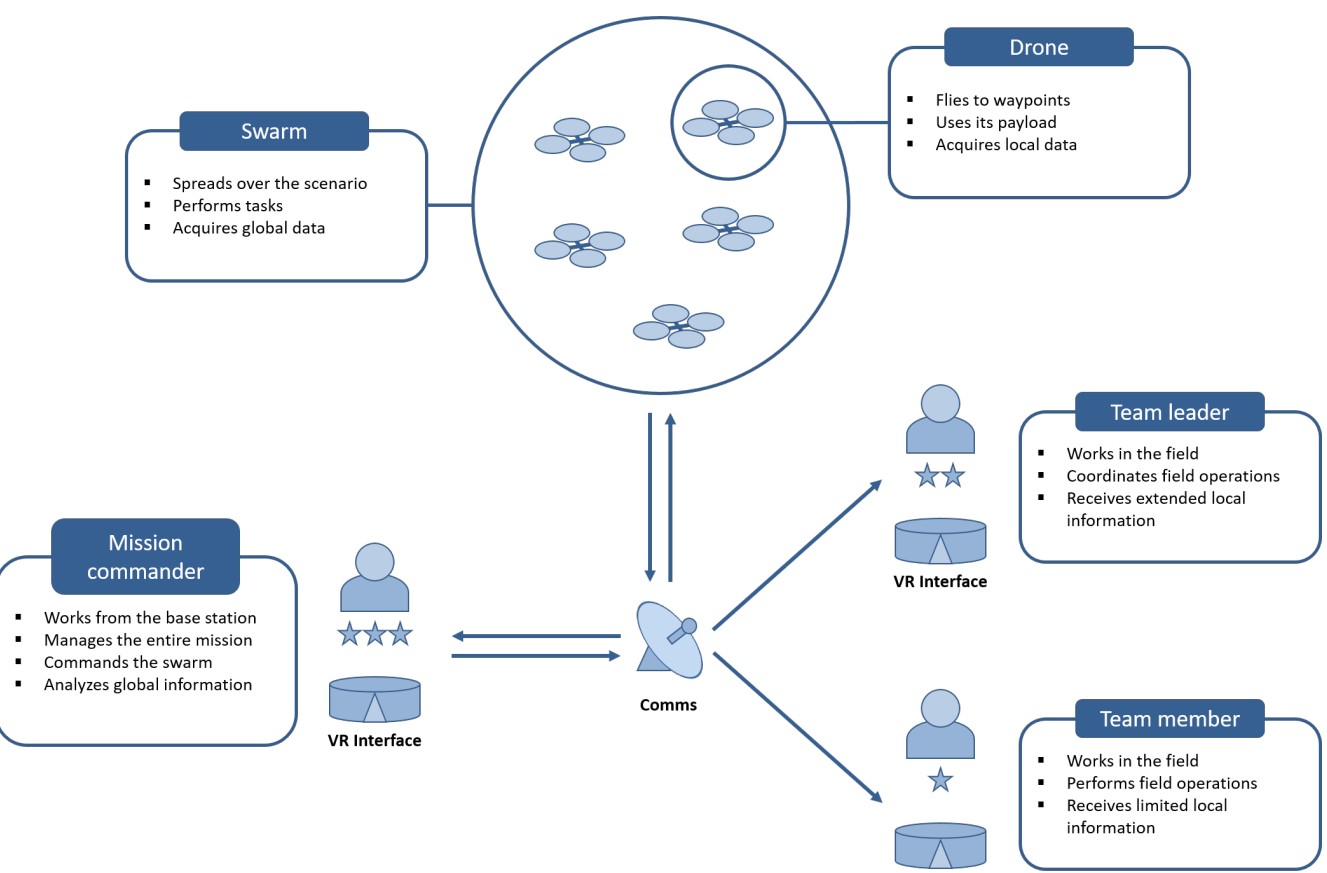

**Figure 3.** System overview.

### 4.1. Mission

The mission has been designed based on current firefighting operations and including research contributions addressed in Sections 2 and 3, respectively. It considers the tasks that could require the participation of the drone swarm, but excludes aerial extinguishing because it would need other types of drones currently in development.

- Prevention: This phase groups the tasks that seek to avoid fires from occurring and control their spread.

  Vegetation mapping: In this task, the drones fly over an area of interest to take ground pictures and build a vegetation map. The number of drones, flight pattern and altitude, and other variables can be tuned to efficiently cover the area and obtain high-quality images. The drones must integrate conventional and multispectral cameras to perform this task. The base station processes images, build a mosaic, detect trees and plants, and recommend actions to the firefighters.

  Fire investigation: This task is developed after the fire is detected. The objective is to find evidence to identify and pursue the perpetrators of the fire. For this purpose, the drones must search around the fire to detect suspicious people, objects, and situations, monitoring static targets, and tracking mobile targets. Although this task is performed after the fire has occurred and the drones have detected it, it is considered a prevention task because it can prevent the occurrence of more outbreaks of the fire. In practice, few drones can perform this task while the rest are carrying out extinguishing tasks.

- Surveillance: This phase considers the tasks that seek to detect fires and alarm firefighting teams early.

  Risk mapping: This task is very similar to vegetation mapping, but creating a map with the risk of fire. This map is useful to know in which areas there is more probability of fire and reinforce surveillance over them. The drones must be equipped with conventional and thermal cameras to perform this task.

  Fire surveillance: In this task, the drones fly over an area of interest looking for potential fires. When one of the drones detects a possible fire, this or another drone must fly closer to check it. For this purpose, the drones must integrate conventional and thermal cameras, as well as environmental sensors: temperature, humidity, and concentrations of combustion gases.

- Extinguishing: This phase groups the task aimed at extinguishing fires and supporting firefighters.

  Fire monitoring: This task is performed to collect information about the fire while the teams on the ground extinguish it. Spatial and temporal information is useful to know the outline of the fire, locate new sources, and predict its evolution. For this purpose, the drones must fly around the fire to incorporate new information from the periphery while keeping updated information from the center. This task needs the same equipment in the drones as risk mapping and fire surveillance.

  Firefighter support: This task aims at supporting the firefighters that are working on the ground to extinguish the fire. For this purpose, the drones must fly around the firefighting teams to collect data about their surroundings and recommend them safe paths and effective actions. Additionally, the drones can transport light resources to firefighters, such as communication devices and protection equipment.

### 4.2. Drone Swarm

The mission described in the previous section can be addressed by several types of aerial robot systems. The first approach is using a heterogeneous drone fleet, so different types of drones can adapt to different types of tasks, increasing the efficiency of the whole mission. For instance, fixed-wing drones can do the tasks that require covering large areas

as surveillance and mapping, whereas rotary-wing drones can do the tasks that require stationary flights as monitoring and support. However, our proposal involves the use of a homogeneous drone swarm to solve this mission. This system relies on the cooperation between drones to accomplish the tasks and not on the adaptation of them to specific tasks. In this case, the same type of drones can perform surveillance and monitoring, but in a different number.

Both systems have advantages and disadvantages in the defined scenario. As already mentioned, heterogeneous fleets can optimize the missions by allocating their different resources to different tasks. Additionally, these systems are easier to control because the drones have more capabilities and need less coordination. On the other hand, drone swarms are more scalable and have more flexibility to adapt to the changes in the scenario. Besides, these systems have better fault tolerance because they can recover from losing one or more members.

We contemplate the definition of robot swarm drawn from [53]: a robot swarm is a group of simple robots, which individually can only perform rudimentary actions, but collectively form an intelligent system and can perform complex tasks. Therefore, we consider a quadcopter fleet as a robot swarm when the fleet consists of a dozen robots, single robots cannot cover the target scenarios, and individual robots are not able to perform the considered tasks. Firefighting missions involve large and complex terrains, where single quadcopters can only collect local information and perform simple actions.

The quadcopters considered for this application shall have the following features:

- Size and weight: No more than $1600 \times 1600 \times 800$ mm unfolded and 15 kg including drone and payload.
- Autonomy: A minimum of 30 min of flight.
- Navigation: Fusion of IMU measurements, visual odometry and GPS/GLONASS/GALILEO signal.
- Control: Capability of reaching and hovering on waypoints.
- Communications: Telemetry and video links in a range of 5 km.
- Payload: Conventional, thermal, and multispectral cameras, as well as temperature, humidity, and gas sensors.

The size and weight were established looking for a compromise between versatility and load capacity. On the one hand, the drones must be light enough to be transported to the fire area in a vehicle and deployed in the field by a person. On the other hand, they must carry up to three cameras, environmental sensors, and communication devices. Finally, we have taken into account the impact of these parameters on flight range and maneuverability. Furthermore, autonomy is an essential aspect of the system: practically, the longer the flight time of the drones, the better the viability of the system in real missions. Current high-performance commercial drones offer around 30 min of continuous flight, but this figure may increase in the following years.

The navigation capabilities of the drones are another relevant aspect of the operation of the system. We have chosen to combine multiple sources to get high accuracy and fault tolerance. Specifically, we consider a high-performance IMU to provide linear acceleration, rotation speed, and orientation, as well as a GNSS receiver to obtain the position, velocity, and time with high frequency. Additionally, on-board cameras can get terrain features, which allow estimating drone motion. Multiple models can integrate the data provided by these sources to obtain the accurate location of the drone, such as Kalman [54] and particle filters [55]. In this way, the drones can preserve enough autonomy to perform their tasks even in GNSS denied or limited environments.

Finally, communications are often a challenge to apply drones in large and distant scenarios. In fire fighting missions, there must be a continuous exchange of information between the different agents: data from the drones to the base station, commands from the base station to the drones, information from the base station to the firefighters, etc. Our proposal to maintain these communications during the missions is to use the vehicles and

robots involved in them as communications relays. However, we estimate that the drones must have a communication range of 5 km to enable this system in the considered scenarios.

As shown in Figure 3, each quadcopter is only able to fly to waypoints and use its payload, whereas the whole fleet can spread over the scenario and perform the required tasks. For instance, a quadcopter can move through a list of waypoints taking images of the terrain, whereas the fleet can cover the whole area monitoring the evolution of the fire. It is made possible thanks the control and coordination algorithms executed by the drones, which allow them to make individual decisions based on local data that produce collective behaviors to perform global tasks. The most representative are behavior-based algorithms, whose efficiency has been validated for surveillance, search, and monitoring tasks in previous works [53,56,57].

Behavior-based algorithms usually consist of multiple behaviors, which process the information and generate possible actions following different patterns, and decision-making module, which fuses the outputs of them and computes the final action. Some common behaviors are inspired in nature, such as "keep distance" and "keep velocity", which are followed by birds' flocks and fishes' shoals. However, some others are devoted to solving specific robot tasks, such as search and surveillance. In both cases, the behaviors have multiple parameters that can be tuned to adapt them to different scenarios.

The drone swarm shall perform the following generic tasks partially drawn from [58]:

- Search: This task involves flying over an area of interest to find some targets, covering every point in that area at least once.
- Surveillance: This task involves flying over an area of interest to find some targets, covering every point multiple times to get updated data.
- Reconnaissance: This task involves flying to a list of points of interest to acquire data.
- Mapping: This task involves flying over an area of interest to build a map, covering every point once to acquire images or data.
- Monitoring: This task involves flying over an event of interest to acquire data.
- Support: This task involves flying over teams that work on the ground to provide them with information about their environment.
- Tracking: This task involves following a mobile target to acquire information or control it.
- Transport: This task involves taking a load from one point to another.

These generic tasks can be used individually or in combination to represent the specific tasks of firefighting missions described above. For instance, fire surveillance can be represented as a combination of surveillance and reconnaissance having fires as targets. The specific tasks and their corresponding generic tasks are collected in Table 1.

**Table 1.** List of tasks considered for firefighting missions.

| Missions | Specific Tasks | Generic Tasks |
| --- | --- | --- |
| Prevention | Vegetation mapping | Mapping |
| | Fire investigation | Search, Monitoring, Tracking |
| Surveillance | Risk mapping | Mapping |
| | Fire surveillance | Surveillance, Reconnaissance |
| Extinguishing | Fire monitoring | Monitoring, Search |
| | Firefighter support | Support, Transport |

*4.3. Team*

Regarding the crew, we consider three principal roles: mission commander, team leaders, and team members. There can be other roles according to the mission and scenario, such as analysts, maintenance workers, communications technicians, etc. As shown in Figure 3 and described below, each role entails different functions, access to information, workplace, and available actions.

- Mission commander: They monitor and controls the mission from the base station, which does not have to be in the fire scenario. All the data collected by the drones is received in the base station and processed to obtain valuable information. Therefore, the mission commander has access to full information on the mission, including the telemetries of drones and measurements on the fire. They must use this information to manage the mission, coordinating the teams on the ground and commanding the swarm. The drone swarm is controlled through high-level commands (e.g., defining areas of interest, variables that must be measured, and required tasks) instead of through low-level orders (e.g., sending specific waypoints and actions to specific drones). This feature is one of the most remarkable strengths of robot swarms, which can self configure to accomplish tasks most accurately, efficiently, and safely. Finally, the mission commander communicates with the team leaders to deliver high-level orders for their teams, establishing the areas where they must work, the tasks that they must perform, and the resources that they can use.
- Team leader: They work in the fire scenario, preferably in a facility or vehicle to ensure communications with the base station. The task of a team leader is to coordinate the field operations of a firefighting team. For this purpose, they receive high-level orders from the mission commander (e.g., area of work and tasks to be performed) and sends low-level commands to the team (e.g., move along a path and attack some flames). In this role, local information is managed both geographically and functionally, that is, the events that happen in the work area and affect the performed tasks.
- Team members: They work in the fire scenario, executing prevention, surveillance, and extinguishing tasks. For this purpose, they can exercise their workforce or use different types of vehicles and machinery. They have access to limited local information, mainly related to the paths that must follow and the actions that must perform. The amount of information should be limited to avoid distractions, but should be enough to ensure their safety.

*4.4. Infrastructure*

A minimal infrastructure is required for the operation of the system. This infrastructure consists of multiple elements that sustain the autonomy of the swarm, enable the communications among the agents, and allow the human-swarm interaction.

As mentioned above, autonomy is a major challenge for applying drone swarms to firefighting missions. Some of the tasks imply continuous flights over target areas, such as fire surveillance and monitoring, whereas some others require a rapid deployment there, such as fire investigation and firefighter support. Therefore, the drones must be able to charge their batteries in the scenario to increase their availability during the missions. For this purpose, charging stations can be distributed throughout the scenario, even using the ground vehicles involved in the mission.

Adaptive and immersive interfaces can improve the situational awareness and reduce the workload of operators in the considered mission. These results have been validated in similar missions, such as the control of multiple robots to perform complex missions [59] and the analysis of the information collected by a drone swarm from a smart city [57].

These interfaces adapt their displays to the mission state and operator preferences, in order to reduce the amount of information and the workload of operator. For this purpose, they can integrate mission and operator models. The first ones allow following the state of the mission and selecting the relevant information according to it, whereas the second ones allow adapting the interface to the operator preferences. The adaptation can be performed through artificial intelligence models like neural networks.

These interfaces apply immersive technologies like virtual reality (VR), augmented reality (AR), and mixed reality (MR) to introduce the operator in the scenario, improving their perception of the environment where the robots are working. VR reproduces virtual environments and allows interacting with their elements; AR enhances real environments

with virtual elements with which the operator can interact, and MR combines real and virtual elements and allows interacting with them [60].

In this work, we consider VR interfaces for the mission commander and AR interfaces for team leaders and members. The mission commander works away from the scenario, so they can focus on the information from the mission. A VR interface can reproduce the scenario, incorporating the real-time information of the swarm and its environment, allowing the operator to move around the scene searching for the best point of view. Meanwhile, team leaders and members work in the scenario, so they must pay most of their attention to the mission. In this case, an AR interface can provide them with relevant information about the mission while keeping their attention in their environment.

## 5. Conclusions

This paper analyzes the current state of firefighting missions and potential technologies that can be applied in the future. To this end, we have conducted two surveys of firefighters to know the main problems they face in their work and their point of view on possible technological solutions. According to the results, the most common problems are the lack of human and material resources for all the activities and the need for real-time information about the evolution of fires during extinguishing tasks. The proposed technologies are positively evaluated when they support their tasks and do not threaten their jobs. Specifically, firefighters support the use of drones as a tool to collect relevant information for prevention, surveillance, and extinguishing activities. In the cases of prevention and surveillance, they approve the generation of maps that help to organize the tasks for preparing vegetation and detect the areas with the highest risk of fires, respectively. In the case of extinguishing, they consider that drones can provide them with real-time information about fires to make their actions safer and more effective.

A review of the literature has been developed to find proposals of robotic systems applied to firefighting tasks. In the case of prevention, there are no proposals for robotizing the vegetation preparation tasks, but there are some developments in the context of forestry and agriculture applicable to them. Conversely, there are multiple proposals for robotizing surveillance tasks, including homogeneous and heterogeneous fleets of drones equipped with conventional, multispectral, and thermal cameras. Finally, in the case of extinguishing, there are multiple approaches to put out fires using autonomous drones, but less to use them to support the firefighters working on the ground.

This paper proposes a concept of operation to apply drone swarms to support fire prevention, surveillance, and extinguishing activities. It considers a fleet of homogeneous quadcopters that individually are only able to visit waypoints and use payloads, but collectively can perform tasks of search, surveillance, reconnaissance, mapping, monitoring, support, tracking, and transport. Three operator roles are defined: mission commander, who commands the swarm and coordinates the mission; team leaders, who coordinate a team on the ground; and team members, who perform the tasks in different areas. These operators have access to different levels of information on the mission through virtual and augmented reality interfaces. On the one hand, this system addresses some of the problems of current operations reported by the firefighters in our survey. It provides the professionals with enhanced information of the scenarios, having an impact on the efficiency of some tasks (e.g., vegetation preparation and fire surveillance) and the safety of some others (e.g., fire extinguishing). On the other hand, some challenges must be overcome, such as the scalability of the system, the training of operators, and the current limitations in the autonomy and communications of drones.

In future works, we are going to develop a complete simulation prototype of the system, as well as a minimum viable product (MVP) with real drones, in order to design, develop and validate the required algorithms.

**Author Contributions:** J.J.R.-G.: Conceptualization, Data curation, Formal analysis, Investigation, Methodology, Project administration, Writing—original draft, Writing—review & editing; A.B.: Funding acquisition, Writing—original draft, Writing—review & editing; E.G.-G.: Writing—original

draft, Investigation, Writing—review & editing. All authors have read and agreed to the published version of the manuscript.

**Funding:** This research received no external funding.

**Institutional Review Board Statement:** Not applicable.

**Informed Consent Statement:** Not applicable.

**Data Availability Statement:** Statistical results are contained within the article. Raw data are not publicly available due to data protection regulation.

**Acknowledgments:** This work has been partially developed within the "UAM Emprende" initiative of the "Universidad Autónoma de Madrid". The authors would like to thank the collaboration of the people involved in this initiative and especially Carlos Romero Moreno. Last but not least, the authors would like to express their gratitude to all the professionals that have taken part in the surveys and interviews. This paper would not have been possible without their knowledge and commitment.

**Conflicts of Interest:** The authors declare no conflict of interest.

## Abbreviations

The following abbreviations are used in this manuscript:

| | |
|---|---|
| AR | Augmented Reality |
| ESA | European Space Agency |
| GCS | Ground Control Station |
| GLONASS | Global'naya Navigatsionnaya Sputnikovaya Sistema |
| GNC | Guidance, Navigation and Control |
| GNSS | Global Navigation Satellite System |
| GPS | Global Positioning System |
| IMU | Inertial Measurement Unit |
| LIDAR | Light Detection and Ranging |
| MR | Mixed Reality |
| MRS | Multi-Robot System |
| MVP | Minimum Viable Product |
| NASA | National Aeronautics and Space Administration |
| UAV | Unmanned Aerial Vehicle |
| UGV | Unmanned Ground Vehicle |
| VR | Virtual Reality |

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
