# Peer review of "A Survey on Robotic Technologies for Forest Firefighting: Applying Drone Swarms to Improve Firefighters’ Efficiency and Safety"

_applsci, doi:10.3390/app11010363_

Round 1

Reviewer 1 Report

Please see attached document called "Referee Report.docx"

Author Response

First of all, we would like to thank you for your work reviewing our paper. We think you have contributed to significantly improving its quality. 

Please find our answers in bold throughout your comments.

I found this article very interesting. It is clearly written and would be very informative for nonexperts and provide readily available tools to experts already involved in firefighting. I found it very appropriate that the authors have conducted two surveys of firefighters' opinions as basis for their proposed solutions. This endows their proposals with a very practical perspective that makes them more likely to be adopted. The results of these surveys are very clearly presented in Figures 1 and 2. The results of these surveys are rather to be expected in that the workers do not readily envision modifying their present manner of doing things. Nevertheless, they are very valuable in showing the nuances in the workers' evaluations of the various technologies. I appreciated the authors' good knowledge of vision systems, as shown in Section 3.2, Lines 233

Thanks for your feedback.

I propose that the authors modify their title because 1) the article is not "a survey on drone swarms", but rather a survey on more general robotic solutions for firefighting 2) the way the title is phrased, means that the "potential robotic solutions" would serve to improve the efficiency and safety of drone swarms. This is not the purpose it is rather to improve the efficiency and safety of firefighters. Indeed, at Line 36, the authors state "This paper analyzes the potential of robotic technologies ..." 3) the title should mention "forest firefighting" instead of simply "firefighting" because the article does not cover all circumstances of firefighting, but rather only pertains to forest fires.

We agree with your comments about the title. We have changed it to "A survey on robotics technologies for forest firefighting: applying drone swarms to improve firefighters' efficiency and safety". In this way, we say that the survey is on "robotic technologies" instead of "drone swarms", mention that the problem is "forest firefighting", and point out that "efficiency and safety" are related to firefighters' work.

The authors should mention at the beginning of the introduction that they will be talking about forest fires.

Again, we find this comment very appropriate. Therefore, we have replaced "fires" with "forest fires" in the abstract, the first sentence of the paper, and multiple sentences in the introduction.

At Line 31: "... to put out them quickly..." should be replaced by "... to put them out quickly..." At Line 91, I believe that the sentence that starts with "Therefore, technological solutions can reduce...." should rather start by something as "Therefore, it would be good if technological solutions could reduce..." 2 At Line 116, the authors mention as a malpractice "smoke or throw glasses in the bush". It is not clear to me how throwing glasses in the bush can cause forest fires. At Line 142 and at other following places, the authors talk about evaluations that are positive, neutral and negative. However, the abscissa of the graphs representing the results in Figure 2 contains only positive values. Therefore, the authors should say what they mean by "neutral and negative" evaluations. At Line 308, the word "difficult" that is incorrectly used as a verb should be replaced by "preventing" or "controlling". At Line 366, the sentence that starts with "It is possible thanks to the ..." should be replaced by "It is made possible thanks to the..." At Line 407, the word "labor" should be replaced by "task".

We have addressed all these issues. You can find them highlighted in the article. More specifically:

  • We have replaced "glasses" with "glass bottles" to avoid the misunderstood. Glass bottles can cause fires by concentrating solar radiation on flammable material.
  • We have explained that "positive" means "high" or "very high" utility of technology, "negative" means "low" or "very low" utility of technology, and "neutral" means "average" utility of technology, using the ratings presented in the firefighters' survey.

Reviewer 2 Report

The paper is a survey paper on drone swarms and the authors discuss potential solutions to improve the overall efficiency. The paper is well structed and easy to follow. The authors discuss adequately the literature methods and also propose a system for firefighting tasks. The paper can be accepted in the present form. 

Author Response

We would like to thank you for reviewing our paper.

Reviewer 3 Report

The article is clear and well written and the literature search is extensive.

The center of the article is the presentation of the survey made to the fire brigade operators and of the bibliographic research conducted on the use of robots in the prevention, surveillance and extinguishing of fires.
The part concerning the robotic system proposal is in my opinion very discursive and qualitative.
I do not see any new elements proposed and no comparison with the actual feasibility of using a complex system in an area where simplicity of intervention is a priority.

Author Response

First of all, thank you for reviewing our article. We think that your review has contributed to improving the quality of our paper. 

Please find our answers in bold throughout your comments.

The article is clear and well written and the literature search is extensive. The center of the article is the presentation of the survey made to the fire brigade operators and of the bibliographic research conducted on the use of robots in the prevention, surveillance and extinguishing of fires.

Thank you for your feedback. 

The part concerning the robotic system proposal is in my opinion very discursive and qualitative. I do not see any new elements proposed and no comparison with the actual feasibility of using a complex system in an area where simplicity of intervention is a priority.

As you said in your previous comment, we focused our article on the current state of firefighting and potential robotic systems to solve its problems. As an addition to this center topics, we aimed to present a concept of operation for applying drone swarms in firefighting missions to open a discussion in the community. For this reason, our proposal is deliberatively general and qualitative and takes many elements from previous research works. However, we found your comment about the feasibility of the system very appropriate, so we have added a discussion about it to the section.

Reviewer 4 Report

The paper surveys the areas of firefighting and presents a framework for utilizing a drone swarm in firefighting missions

Major comments

-The survey included a few general questions about potential use of robotics in this area which is not enough to develop a good understanding of the research problem.

-The methodology to build the two survey questioners is not convincing and how they were validated is not presented. Furthermore, survey results are displayed with no analysis of their significance and correctness.  

-The identified features for swarms in section “4.2. Drone swarm” are not justified

-Additionally, in the presented framework for utilizing a drone swarm in firefighting missions, it is not clear why the UAVs should work as a homogeneous swarm. The diversity of tasks that should be handled by them calls for a heterogeneous multi-UAV team rather than a drone swarm.

Minor comments

-Paragraph started at line 23 stated information without a reference. 

- More information about control and coordination algorithms should be added.

Author Response

First of all, we would like to thank you for your work reviewing our paper. We think you have contributed to significantly improving its quality.

Please find our answers in bold throughout your comments.

The paper surveys the areas of firefighting and presents a framework for utilizing a drone swarm in firefighting missions

Major comments

-The survey included a few general questions about potential use of robotics in this area which is not enough to develop a good understanding of the research problem.

We designed the technology survey taking into account the following premises:

  • It must not have more than 15 questions or require more than 5 minutes to reach the desired answer ratio.
  • It must ask about the target technologies (drones and immersive interfaces) and their main alternatives (satellite, cameras, mobile devices...) to avoid introducing bias in the participants.
  • It must be easy to understand for firefighting professionals, who may not know about robotics, automation...
  • It should present the technologies as a support and not as a replacement for the workers to avoid bias in their evaluation.

In our opinion, we met all these conditions except partially the fourth, but this fact is mentioned in the article.

We would have liked to ask for more robotic technologies, but we had to prioritize the success of the survey.

We have added this information to the paper just before the subsections devoted to problem and technology surveys. 

-The methodology to build the two survey questioners is not convincing and how they were validated is not presented. Furthermore, survey results are displayed with no analysis of their significance and correctness.  

Our objective was to collect opinions from firefighting professionals about the problems in their work and potential technological solutions. We wanted this information to analyze which technologies can be useful for them and then formulate our proposal. For this purpose, we performed this survey to involve a high number of professionals and avoid overestimating individual opinions. Later, we did some interviews with selected participants to obtain more detailed information about their answers. In summary, we did not intend to do sociological or demographic research but collect relevant opinions about these topics.

We designed the survey and its dissemination taking into account the following premises:

  • As mentioned above, we designed the surveys to last no more than 5 minutes to obtain an acceptable answer ratio.
  • We separated the surveys of problems and technologies to avoid the potential influence of the questions of the second one on the answers of the first one.
  • We asked not only about target technologies but also about potential alternatives, searching to avoid introducing a bias towards drones.
  • We sent the first survey by email to fire stations and firefighter unions, and we shared it in groups on various social networks, in order to get a sample that includes all the firefighting roles in all the regions of Spain.  

We consider that the sample meets these requirements because it includes all the possible roles with enough participants. Unfortunately, there is no official information on the actual number of professionals with each role.

We have published the raw results of the survey because of two reasons: we considered that it is enough for our study about current problems of firefighting and potential technological solutions, and we wanted to share this information with every interested party. The processing of this data considering the differences between the sample and the actual composition of firefighter corps is not within the scope of our work. 

We have added this information to the paper in the corresponding sections.

If you still have doubts about the methodology, please explain which specific aspects we should improve.

-The identified features for swarms in section “4.2. Drone swarm” are not justified

We have added some paragraphs that justify the selected features of the drones. In general, we choose these parameters after analyzing the mission requirements and drones available at the market.

-Additionally, in the presented framework for utilizing a drone swarm in firefighting missions, it is not clear why the UAVs should work as a homogeneous swarm. The diversity of tasks that should be handled by them calls for a heterogeneous multi-UAV team rather than a drone swarm.

A heterogeneous multi-robot system and a homogeneous robot swarm are two valid strategies to solve this mission. The first one relies on the use of different types of drones that adapt to the different types of tasks. The second one relies on the cooperation among the drones to accomplish the tasks. In this project, we are researching the potential of homogeneous drone swarms to perform complex missions, so our proposal follows this research line. We have added a paragraph explaining that different multi-robot systems can be used to solve this mission. 

Minor comments

-Paragraph started at line 23 stated information without a reference. 

If you mean the paragraph that starts with "Quantifying the fires and their consequences…", all the information is from the World Fire Statistics, whose reference is provided in the text.

- More information about control and coordination algorithms should be added.

We have added some paragraphs explaining the work of behavior-based algorithms.

Round 2

Reviewer 4 Report

The authors have not adequately addressed any of our concerns.

Author Response

Please find our answers throughout your comments and highlighted in bold.

The authors need to carefully read and understand each comment before starting any response. Below, we include our previous comments and, in blue, we provide more details about each comment for the authors to develop better understanding of how to respond, or improve the manuscript, based on the comments:

First of all, we would like to thank the editor for addressing this situation and the reviewer for broadening the comments. We agree that there has been a misunderstanding, but we disagree with the cause. We read the first-round comments carefully, but we found some of them too vague. The reviewer guidelines of this journal say: "Broad comments highlighting areas of strength and weakness. These comments should be specific enough for authors to be able to respond". We honestly think this was not the case. 

Major comments

-The survey included a few general questions about potential use of robotics in this area which is not enough to develop a good understanding of the research problem.

What is the research question and objectives? How are question of these two surveys related to this question? Usually, there should be a direct mapping between each survey question or group of questions and a research objective. The authors need to explain this mapping as it is currently unclear.

The research questions of our study are 1) "What are the main problems in current firefighting operations?", and 2) "How can technology contribute to solving them?".

Both objectives are mentioned more or less explicitly througout the text: e.g., "This work presents a complete survey of firefighters on the current problems in their work and the potential technological solutions" (Abstract), "it describes current operations of fire prevention, surveillance, and extinguishing, collects relevant statistics to identify main problems, and presents the opinions of professionals through two surveys" (Section 2), "These activities aimed to collect information about current problems of firefighting and opinions about potential technological solutions" (Section 2), and so on.

However, we have rewritten some paragraphs of the introduction to directly mention these research questions.  

The first survey is directly related to the first research question. The objective of this survey was to collect opinions about current problems in firefighting operations. For this purpose, we split firefighting missions into prevention, surveillance, and extinguishing tasks, and then formulate a couple of questions about each of these tasks.

S1Q1-Importance of prevention tasks: In the review of the literature, we found two different prevention strategies: social (awareness campaigns and prosecution of suspects), and environmental (management of land and vegetation in forests). We wanted to know the importance that professionals give to these strategies. In this way, we can start estimating if robots can improve current operations, given that they can perform a role in environmental but not in social strategies.

S1Q2-Main problems in prevention tasks: This second question directly asks for the problems in prevention tasks, which is without any doubt related to the first research question. As above mentioned, we defined the options after studying the literature and with the advice of firefighting professionals. 

S1Q3-Importance of surveillance means: In the review of literature, we found that means are more interesting than tasks for surveillance. Therefore, we wanted to compare the official data with the professionals' opinions in the context of fire surveillance. This question allowed us to know their opinion on the drone systems that are currently used in surveillance tasks.

S1Q4-Main problems in surveillance tasks: Same as S1Q2 but with surveillance tasks. 

S1Q5-Main problems in extinguishing tasks: Same as S1Q2 and S1Q4 but with extinguishing tasks. 

The second survey is related to the second research question. We included the target technologies of our study, which are drone swarms and immersive interfaces. Additionally, we included some control technologies that allowed us to compare the evaluations of target technologies, such as incentive systems, satellite imagery, video surveillance systems, mobile devices, and voice assistants. We selected these technologies after a search in research and commercial literature related to firefighting technologies. We know that our selection is not exhaustive and that some relevant technologies are missing. However, we prioritize the control of survey length and duration to maximize the answer ratio. 

S2Q1-Incentives for farmers/ranchers to avoid the use of fire: There are many proposals in this direction and we should include a technology related to fire prevention through social activities.

S2Q2-Satellite images to support the preparation of vegetation: This technology is the main competitor of drones in agricultural and environmental applications.

S2Q3-Drone images to support the preparation of vegetation: This is a target technology in our project.

S2Q4-Camera system for early fire detection: These systems are widely used in security applications and there are some proposals for fire detection. We consider them as the main competitors of drones for fire surveillance.

S2Q5-Autonomous drones for early fire detection: This is a target technology in our project.

S2Q6-Artificial intelligence to predict the risk of fire: This is a target technology in our project. We wanted to separate the concepts of drones and artificial intelligence to know the opinions about them.

S2Q7-Autonomous drones to monitor the evolution of fires: This is a target technology in our project.

S2Q8-Augmented reality to receive information during operations: This is a target technology in our project.

S2Q9-Mobile device to receive information during operations: This is an alternative of immersive interfaces that firefighters often use.

S2Q10-Voice assistant to receive information during operations: This is an alternative of immersive interfaces that firefighters often use.

We have added some paragraphs explaining the question choice for each one of the surveys. You can find them at the beginning of the subsections devoted to the surveys. 

We hope this explanation is enough to solve your concern. 

-The methodology to build the two survey questioners is not convincing and how they were validated is not presented. Furthermore, survey results are displayed with no analysis of their significance and correctness.

We are going answer both questions separately because they are different.

There are special techniques that are usually used by scientists to validate a survey before using it to ensure that the items can be understood and correctly interpreted by the intended respondents. So, the question here is have the authors validated their questioners? How?

We made the following actions to validate our questionnaires:

1) We had the support of several firefighters in the development of the questionnaires. They gave us advice when writing the questions and possible answers. 

2) We sent the questionnaires to a sample of twenty professionals before their dissemination. Thereby, we could analyze their answers to check their coherence. 

3) We allowed open answers to some questions so the professionals could make their contributions. 

4) We gave our contact data in both surveys so the respondents could contact us if they had any doubt.

In this way, we can state that both surveys were understandable by our target public. 

We have added the missing part of this information to the article.

Additionally, there are special techniques that are usually used by scientists to measure the significance and correctness of survey results. Have the authors evaluated the significance and correctness of their results? How?

First of all, we would like to clarify the objectives of our work. We conducted both surveys to gather opinions from professionals about their problems and possible technological solutions. These surveys allowed us to obtain more information more efficiently than interviews. 

We did not intend to study the global population of firefighters in Spain. In fact, if you read carefully Sections 2.1 and 2.2, we present the results with sentences like "the professionals surveyed assign..." or "the participants consider...", instead of "Spanish firefighters assign..." or "firefighting professionals consider...". Therefore, we do not need to ensure the representativeness of the sample.

However, we gave evidence that our sample can be representative. The sample includes all the relevant roles in forest firefighting missions in Spain: forest firefighter, technician, firefighter, pilot, forest agent, and volunteer. This can be checked by reading official reports (https://www.mapa.gob.es/es/desarrollo-rural/estadisticas/incendios-decenio-2006-2015_tcm30-511095.pdf). Unfortunately, official reports do not quantify the total number of professionals in each role, given that they work for different administrations (national, regional, and local) and some of them perform seasonal or temporary works. Therefore, we can state that our sample has all the relevant roles in a minimum number, but we cannot know if they are in the same percentage as in the whole population. 

Regarding the significance and correctness of the results, first, we would like to analyze the surveys in more detail. Two relevant topics are how we collected data and how we presented results. 

Survey 1: We asked for the importance of different topics: tasks, means, and problems. The respondents had to evaluate it between "less important" and "very important" with three middle terms. Therefore, we consider the responses are qualitative (i.e., an adjective) instead of quantitative (i.e., a rating). Additionally, there may be issues when translating from the original survey language to the paper language. We presented the average score between 1 and 5 because it required minimum processing, and we found it easy to understand. However, we want to remark that the answers were qualitative instead of quantitative. 

Survey 2: We asked for the usefulness of different technologies. The respondents had to evaluate them between "very low" and "very high" utility with three middle terms. Again, we consider the responses are qualitative instead of quantitative. In this case, we presented the percentages of the different answers because we found some interesting nuances: e.g., a technology can receive more extreme support and rejection than another, while the two technologies have the same rating. Again, there may be issues when translating from the original survey language to the paper language. 

Taking into account this information, we consider that common statistical tests are not necessary nor applicable. Let's see some examples: If one technology has 70% acceptance and 30% rejection in the survey, these percentages are enough to say that this technology is accepted by the respondents. If one technology has 60% acceptance and another 40% acceptance in the survey, these percentages are enough to say that the first technology is more accepted than the second one. 

As we said above, we do not intend to study the global population of firefighters in Spain nor to find differences in their opinions according to their role. In this case, there would be multiple statistical hypothesis tests that we could apply. 

If you know a statistical test that can be applied in this context, please give us the name so we can apply it. 

The identified features for swarms in section “4.2. Drone swarm” are not justified

The authors should justify the selection of the identified features in section 4.2. For
example, why in size and weight a drone for a firefighting mission should be no more than 1600x1600x800 mm unfolded and 15 kg including drone and 385 payload?

In the first-round of revision, we added three paragraphs justifying the proposed features for the drones, just below the list of features. These features were defined after analyzing the mission requirements and drones available at the market.

For instance, we set the size and weight considering that a vehicle should be able to transport multiple drones, and a firefighter to carry one of them. 

Did you read these paragraphs? What information is missing there?

-Additionally, in the presented framework for utilizing a drone swarm in firefighting
missions, it is not clear why the UAVs should work as a homogeneous swarm. The
diversity of tasks that should be handled by them calls for a heterogeneous multi-UAV team rather than a drone swarm.

The authors suggest the swarm should be homogeneous. Why is this necessary or
advantageous for firefighting? What are advantages and disadvantages of other options?

First, we consider that a swarm is necessarily homogeneous. The most common definition of "swarm" in the literature considers "relatively homogeneous robots, all identical or from a few typologies" (http://www.scholarpedia.org/Swarm_intelligence). We follow this definition in the present and previous works.

Then, we choose a homogeneous drone swarm instead of a heterogeneous drone fleet because of two reasons: 1) Our research line involves developing control algorithms for swarms and applying these systems in realistic missions, and 2) we think a drone swarm can accomplish the tasks considered for firefighting missions.

As we said in the revision and the paper, both systems can perform the considered mission. We do not state that the drones should be homogeneous anywhere in our paper. We only say that our proposal is a homogeneous drone swarm and then describe it in more detail. We think this is a design decision that we can make in our project. 

However, as you recommend it and we think it is interesting, we have added a paragraph discussing the advantages and disadvantages of homogeneous and heterogeneous systems in this mission. You can find it at the beginning of Section 4.2.